# Enhanced Adhesion of Copper Films on Fused Silica Glass Substrate by Plasma Pre-Treatment

**DOI:** 10.3390/ma16145152

**Published:** 2023-07-21

**Authors:** Liqing Yang, Xianli Yang, Fei Gao, Yongmao Guan, Rui Wan, Pengfei Wang

**Affiliations:** 1State Key Laboratory of Transient Optics and Photonics, Xi’an Institute of Optics and Precision Mechanics, Chinese Academy of Sciences (CAS), Xi’an 710119, China; yangliqing@opt.ac.cn (L.Y.); gaofei1980@opt.ac.cn (F.G.); guanyongmao@opt.cn (Y.G.); wanrui@opt.cn (R.W.); 2Institute for Agro-Food Standards and Testing Technology, Shanghai Academy of Agricultural Sciences, Shanghai 200062, China; yangxianli@saas.sh.cn

**Keywords:** copper films, plasma pre-treatment, adhesion, diffusion junction layer

## Abstract

A non-thermal atmospheric jet plasma pre-treatment technique was introduced to help the growth of extremely sticky copper films on fused silica glass substrates. A tape test was utilized to assess the bonding quality between copper films and fused silica glass substrates. AFM was used to characterize the sample surface roughness and XPS for chemical bonding characterization. The Owens–Wendt method and a Theta Lite Optical Tensiometer were used to analyze the contact angle and surface energy. The results showed that the surface energy greatly increased from 34.6 ± 0.3 mJ·m^−2^ to 55.9 ± 0.4 mJ·m^−2^ after 25 s plasma pre-treatment due to the increasing Si-O and Si-N concentrations, which brought about the electrostatic force increasing at the copper/glass interface. After 25 s plasma pre-treatment, the average surface roughness (Sa) grew from 0.8 ± 0.1 nm to 2.4 ± 0.3 nm. With higher surface roughness, there were more spaces and vacancies for the copper atoms to make contact on the bonded surfaces and increase the mechanical bite force. The electrostatic force and the mechanical bite force on the interface helped to form an atomic diffusion connection layer and improved the interactions between the copper film and the glass substrate. The findings in the SEM supported the conclusions stated above. Therefore, the adhesion between copper films and fused silica glass substrates increased by about 20% by 25 s plasma pre-treatment compared with the untreated glass substrate.

## 1. Introduction

Copper represents one of the most important conductive materials because of its excellent conductivity, solderability and stability. Copper films on glass substrates as a kind of conductive thin film are widely used in various film integrated circuits [1,2]. It is essential to prepare copper film with a low-cost, high-quality graded growing process when considering for commercial applications. A number of techniques have been developed to grow copper film, including modified activated reactive evaporation (MARE), DC reactive magnetron sputtering, RF reactive sputtering, pulse laser deposition, ion-assisted vapor deposition, and chemical vapor deposition (CVD), etc. [3,4,5]. Meanwhile, glass substrates always exhibit poor adherence to copper films [6,7] due to the poor bonding strength of the metal–glass interface, which is correlated with the coating process parameters, its growth mode, and the structure of the substrates. The most popular method for enhancing the glass substrate’s adhesion ability is to pre-deposit an adhesion layer made of Cr, Ti or TiW, etc. [8,9,10,11]. Normally, expensive vacuum deposition and patterning methods including lithography and etching are used to achieve this technology [12,13]. Furthermore, these methods raise the cost of manufacturing in addition to producing a lot of toxic metals. Additionally, researchers prepared pure copper films on glass substrates via the thermal decomposition of copper-based pastes. For instance, Joo and Lee et al. reported the fabrication of pure copper films on polyamide using a single paste formulation containing copper complexes with good adhesion properties [14,15,16]. Lee fabricated pure copper films strongly adhered on glass substrates by printing a paste containing Cu(COOH)_2_ according to ASTM class 5B [17]; however, this solution needs to go through a lot of preparation procedures (mechanical mixing, physical filtration and wash, then the paste is transported to a heating furnace, etc.), and it is a source of liquid and gaseous wastes. 

Therefore, enhancing the adhesive qualities of the glass substrate through an environmentally friendly procedure like cold plasma pre-treatment could be an intriguing alternative. The traditional ways to generate cold plasma require high-vacuum equipment, making them expensive and difficult to integrate for in-line manufacturing processes. Atmospheric pressure plasma technologies are of lower costs and are suitable for large-scale industrial applications [18,19]. Additionally, this technology is simple to use and has no impact on the optical characteristics of glass prior to the deposition of the functional thin film. In addition, cold plasma pre-treatment is usually performed in the atmosphere for a few seconds or minutes before deposition of a functional coating, which significantly reduces the time and expense of glass treatment [20,21]. More generally, scientists discovered that plasma pre-treatment of glass has a cleaning effect and creates oxygen-based functional groups that enhance the glass substrates’ hydrophilicity. The substrates treated with plasma adhere better as a result of this dual action [22,23,24,25]. Moreover, some scholars proved that electron transfer occurs when metal nanocrystals and silicon-containing elements would be interacted through first-principles calculations and finite element method simulations, resulting in interface performance changes, such as changes in the interface energy and the electric field strength [26,27]. The interesting facts inspire enthusiasm for further research on the mechanism of the integration of metallic copper and fused silica glass interfaces.

In this study, we will present a non-thermal atmospheric jet plasma pre-treatment approach for the development of highly adhesive copper films on the fused silica substrate. Meanwhile, we will investigate the potential mechanism of air cold plasma pre-treatment on fused silica and how it relates to subsequent mechanical adhesion with copper layers.

## 2. Materials and Methods

In this investigation, 200 mm × 100 mm × 2 mm fused silica glasses were employed as the substrates for copper films. The surface roughness was Rq < 0.6 nm as determined via an interferometer over a spatial bandwidth of 1–300 μm. New glass substrates were cleaned in an ultrasonic cleaner for 10–15 min using acetone and ethanol (1:1) and de-ionized (DI) water.

Using pulsed AC power with a voltage of 2 kV and a frequency of 250 kHz, air cold plasma pre-treatments of fused silica substrates were carried out at varied discharge times, as shown in Figure 1 [28]. The following variables were used to vary the plasma pre-treatment conditions: power (2 kW), the distance between the electrode and the glass substrate (2 mm), and the plasma treating period (5, 10, 15, 20 and 25 s). The size of the treatment area was around 8 × 10^−2^ cm^2^ each time. The plasma pre-treatment time was varied to control the moving speed of the substrate. 

Using the traditional Owens–Wendt method, the surface energy of untreated and plasma pre-treated glasses was calculated. With the aid of an Attention Theta Lite Optical Tensiometer and an imaging source camera, contact angle measurements of two different liquids (water and diiodomethane) were taken. Five measurements were taken at five different locations on the sample surface in order to lessen the margin for error.

The XPS spectra were produced utilizing a monochromatized MgK (E = 1253.6 eV) excitation radiation and an Omicron brand EA 125 spectrometer on average. Using a neutralizer pistol to spray low energy Ar^+^ ions all over the sample surface, surface charging was reduced to a minimum. All binding energies were set to the C1s peak at 284.5 eV [21] to account for surface charge. The software XPSPEAK 4.1 was used to decompose the XPS spectra. Small vacuum chambers were used to transfer plasma-pre-treated samples to the XPS apparatus in order to prevent surface evolution following exposure to outside air.

After plasma pre-treatment, copper films were deposited by sputtering a copper target (99.999% purity, 30 mm diameter) in the electron beam assisted evaporation coating instrument. Target to substrate separation remained constant at 150 mm. With a deposition period of 300 s, a DC power density of 3.3 W/cm^2^, an argon flow rate of 20 sccm, and a working pressure of 8 × 10^−3^ mbar, films with a thickness of 200 nm were produced. The film thickness was monitored by a quartz crystal unit and measured by a Talystep profilometer into the evaporation coating instrument. The deposition rate was 10 Å·S^−1^ for deposition of copper. The spectroscopic ellipsometry observations with refractive indices were used to calculate the deposition rates.

Atomic force microscope (AFM, Bruker Dimension Icon Bruker Corportion, Santa Barbara, CA, USA) tapping mode was used to examine the surface morphologies of untreated and plasma pre-treated substrates. The surface roughness had been measured on a size of 5.0 × 5.0 μm^2^.

According to the American Society for Testing and Materials (ASTM) D3359 standard [29], a tape test following cross cuts was performed to assess the bond between fused silica substrates and copper films. 3M^TM^ 2525 and 3M 610 Scotch tapes with adhesion strengths on steel of 75 N/mm and 47 N/mm, respectively, were utilized as the adhesive tapes for the test. Each result was the average of at least six individual measurements. After peeling, we were able to express the adherence in terms of the percentage of copper that was still on the glass substrate by processing optical microscopic pictures of the samples. 

## 3. Results

Here, we examined the surface topography, chemical alterations, contact angle and surface energy, as well as the adhesive qualities of the copper films that had been formed. The goal of the analysis was to determine how the pre-plasma affected the adhesion characteristics of the films. 

### 3.1. Adhesion Testing

Table 1 summarizes the adhesion characteristics of copper films on glass substrates in order to evaluate the effect of plasma pre-treatment on glass. The morphological changes of the copper grids on the glass substrates and the amount of copper films adhering to the tapes were clearly visible in the adhesion experiment in the images taken by the microscope. Copper film shedding was dramatically reduced with the plasma pre-treatment time increasing. Almost no copper film adhered to the tape when the glass substrate was plasma treated for 25 s.

The proportion of copper that was still on the glass surface after the peeling test was calculated as shown in Figure 2, which depicts the adhesion between copper film and glass as a function of plasma pre-treatment time. The remaining copper adheres to the untreated glass with just 78% and 80%, respectively, for the tapes 3M^TM^-2525 and 3M-610. There was no significant change after a 5 s plasma pre-treatment of the glass substrate, but the change was significant after treatment for 10 s. After treatment for 20–25 s, there was a slight increase compared to 10 s. And the optimal performance was achieved at 25 s plasma pre-treatment; the amount of copper on the glass substrate was still 98% for the 3M^TM^-2525 tape and 99% for the 3M-610 tape, which increased by 20% and 19%, respectively. Similar outcomes were obtained by Relyon’s research, which saw a 3.7-fold improvement in average adhesive strength after plasma pre-treatment compared to untreated glass substrates [30].

### 3.2. XPS Analyses

The untreated and atmospheric plasma-treated glasses were analyzed using X-ray photoelectron spectroscopy (XPS) at various treating times. Results showed that the elements present were silicon, oxygen, calcium and the usual carbon contamination. Figure 3 shows the XPS spectrum of the glass after 25 s plasma pre-treatment in which the peaks of Si2p, Si2s, O1s, N1s, Ca2p and C1s were identified. 

The Si2p high-resolution spectra of untreated (as-received) and atmospheric plasma-treated glass samples at a time of 25 s are displayed in Figure 4. Using Gaussian–Lorenztian fitting, the Si2p peak centered at 102.9 eV that corresponded to the as-received glass (Figure 4a) was deconvoluted into four peaks, two peaks from Si2p_3/2_ and another two peaks of the spin orbit splitting, with an energy difference of 1 eV. The non-bridging oxygen Si-O-Si bond was assigned to the Si2p_3/2_ component at 101.9 eV, the bridging oxygen Si-O-Si bond at 102.9 eV, and the Si-N bond at 397.6 eV, respectively. After plasma pre-treatment, the percentage of Si2p dropped from 7.49% to 4.01%, but the percentage of Si-N rose from 0.33% to 0.52% (Figure 4b). These details demonstrated that nitrogen elements were grafted onto the surface of the glass. 

The high-resolution spectra of untreated and atmospheric plasma-treated glass samples at 25 s are displayed in Figure 5. The fitted O1s spectra indicated, as shown in Figure 5a,b, that the surface of the substrate was mostly at 530.9 eV to Si-O (non-bridging oxygen) and at 531.8 eV to Si-O-Si (bridging oxygen). The intensity of O1s rose after 25 s of plasma pre-treatment. Si-O-Si and Si-O both showed an increase in intensity. In particular, Si-O intensity was substantially higher than it was in the untreated sample. O1s ranged in proportion from 15.96% to 20.91%. Therefore, it followed that a rise in hydrophilicity and surface energy would result from strong electronegativity of oxygen, which is shown in Section 3.2.

The percentages of regions of various peaks in the Si-N band, Si2p band and oxygen band are summarized in Table 2. Although the quantity of Si in the sample surface reduced, the nitrogen and oxygen concentrations in the sample grew as the air plasma pre-treatment continued. The findings showed that the plasma pre-treatment process activated the surfaces and increased their reactivity, which was advantageous when combined with the potent electronegativity of oxygen and nitrogen. Additionally, the increase in the concentration of O1s for the treated samples also produced the cleaning effect and created polar oxy-based functional groups on the surfaces [31,32]. These groups were typically linked to the improvement of the adhesive qualities and hydrophilicity of the surfaces [24,33]. 

### 3.3. Contact Angle and Surface Energy

Contact angle and surface energy analyses were carried out in order to study how the substrate’s surface characteristics affected adhesion layers as well as the growth of the plasma treatment time. Table 3 displays the surface energy and contact angle trends for glass samples that had been exposed to atmospheric plasma over time and those that had not. Glass substrate that had been pre-treated with short-duration plasma had much lower contact angles against water and diiodomethane (down from 70.2° to 41.2° and 64.9° to 60.1°, respectively) than untreated glass samples. These glass samples had much higher surface hydrophilicity. The surface energy of glass samples was calculated using the traditional Wendt–Owens method based on contact angle measurements [34]. Total surface energy for the untreated glass sample was 34.6 ± 0.8 mJ·m^−2^. In the case of a 25 s pre-treatment, the total surface energy reached a maximum value of 55.9 ± 0.4 mJ·m^−2^. Furthermore, the polar component of the surface energy, which was mostly affected by the surface pre-treatment, dramatically increased from 16.1 ± 0.3 mJ·m^−2^ to 36.2 ± 0.3 mJ·m^−2^ after plasma pre-treatment. Before stabilizing at roughly 16.7± 0.3 mJ·m^−2^, the dispersive component reduced marginally. According to the interpretation in terms of adhesion, a rise in the number of polar functional groups such as Si-O- (from 2.01% to 3.96%) and Si-N- (from 0.33–0.52%) on the glass surface likely caused more interactions with copper atoms. As a result, at the copper/glass interface, an interfacial fusion layer or interface element diffusion may form. Because the polar component increased, the electric field intensity augmented at the copper/glass interface, that is, the electrostatic force increased between the copper film and the glass substrate, which would inevitably tend to improve the adhesion performance. It was shown that the electric field intensity increasing could increase adhesion at the interface [27]. 

### 3.4. Surface Topography

AFM was used to examine the surface topography of samples that had not been treated and those that had been treated with plasma. The three-dimensional topography of glass substrate samples is shown in Figure 6. 

As demonstrated in Table 4, the surface roughness of glass samples increased with increasing air plasma pre-treatment time. In detail, the roughness value significantly increased from 0.8 ± 0.1 nm to 2.4 ± 0.3 nm when the plasma pre-treatment period was increased to 25 s. From 5 s to 25 s, the RMS roughness value of plasma-pre-treated glass increased to 1.5 ± 0.2 nm and 3.5 ± 0.1 nm, respectively.

In addition, we found a positive association between glass surface roughness and adhesion, with an improvement in the percentage of copper remaining after peeling with increasing glass surface roughness. Therefore, we may assume that the surface roughness did have a direct bearing on the adhesion between copper films and fused silica glass. The plasma pre-treatment had increased the surface roughness, which increased the specific surface area of the substrate and led to a tendency for an atomic expansion connection layer to form between the copper film and the glass substrate, improving the interface layer’s ability to adhere. The findings in the SEM, as presented in Figure 7, supported the conclusions stated above. In Figure 7a, the interface between the glass substrate and the copper layer is clearly defined, and in Figure 7b, the interface between the two is clearly integrated. Similarly, Lili Cao et al. demonstrated that the amount of ordered nanoparticles reduced as the contact angle of the hydrophobic surface increased [3]. Surface roughness is important for enhancing the quality of diffusion bonding joints, according to Zhulei et al. [35]. Additionally, as demonstrated by Frederic [36], surface roughness enhances the quantity of contacts between the substrate and the atoms of the metal layer. The surface roughness increasing meant that a large number of nanoscale fine potholes, bumps or ravines had formed on the surface, which could increase the specific surface area of the substrate. That is to say, with higher surface roughness, there were more spaces and vacancies for the sputtered copper atoms to make contact on the bonded surfaces and increase the mechanical bite force. Mechanical bite force was the interaction force that resulted from the action of two uneven surfaces by contacting, imbedding and interlacing. Therefore, the force per unit area was increased with the surface roughness of the glass substrate, contributing to a higher adhesive strength. 

## 4. Conclusions

In this study, we illustrated the effective improvement on the adhesion between copper film and fused silica glass substrates using the non-thermal atmospheric jet plasma.

An analysis of XPS data indicated that the concentrations of polar functional groups on the glass surface were increased. Following prolonged plasma pre-treatment times, up to 25 s, the percentage of Si2p dropped from 7.49% to 4.01%, but the percentage of Si-N rose from 0.33% to 0.52% and O1s ranged in proportion from 15.96% to 20.91%. The results demonstrated that the plasma pre-treatment process activated the surfaces and increased their reactivity, which was advantageous to increasing the hydrophilicity and the electrostatic force. Moreover, electrostatic force would result in more ordered copper particles on the glass substrate.

The contact angle and surface energy data indicated the plasma pre-treatment for 25 s had a significant effect on the wettability of glass surfaces. The total surface energy rose from 34.6 ± 0.8 mJ·m^−2^ to 55.9 ± 0.4 mJ·m^−2^, while the water contact angles reduced from an initial 70.2° to less than 41.2°. Furthermore, the polar component of the surface energy, which was mostly affected by the surface pre-treatment, dramatically increased from 16.1 ± 0.3 mJ·m^−2^ to 36.2 ± 0.3 mJ·m^−2^ after plasma pre-treatment. As a result, the electric field intensity augmented at the copper/glass interface. Therefore, an interfacial fusion layer or interface element diffusion may form at the copper/glass interface. 

The AFM data showed that the surface roughness (Ra) increased from 0.8 ± 0.1 nm to 2.4 ± 0.3 nm after 25 s plasma pre-treatment, which would result in increasing the specific surface area of the substrate. Moreover, the interactions between the copper film and the glass surface and the mechanical bite force would increase. As a result, an atomic diffusion connection layer formed. The findings were verified by the SEM results, which showed that copper films and the fused quartz glass surface formed an atomic connection layer. Thereby, this improved adhesion performance between the copper layer and the glass substrate. 

The adhesion testing data showed the remaining copper on the untreated glass substrate was approximately 78% and 80% for the tapes 3M-2525 and 3M-610, respectively. After 25 s plasma pre-treatment, the amount of copper on the glass substrate was still 98% for the 3M^TM^-2525 tape and 99% for the 3M-610 tape, which increased by 20% and 19%, respectively. 

As a result, there was a noticeable increase in the adhesion of copper films to the surface of fused quartz glass by air plasma pre-treatment. The primary mechanism was that the plasma pre-treatment caused an increase in the electrostatic force and mechanical bite force at the interface between the glass substrate and copper film, which caused an increase in the ordered copper particles and the creation of a fusion layer. Other adhesion mechanisms, such as the impact of putting an adhesive transition coating on the surface beforehand or surface cleaning, were distinct from this.

## Figures and Tables

**Figure 1 materials-16-05152-f001:**
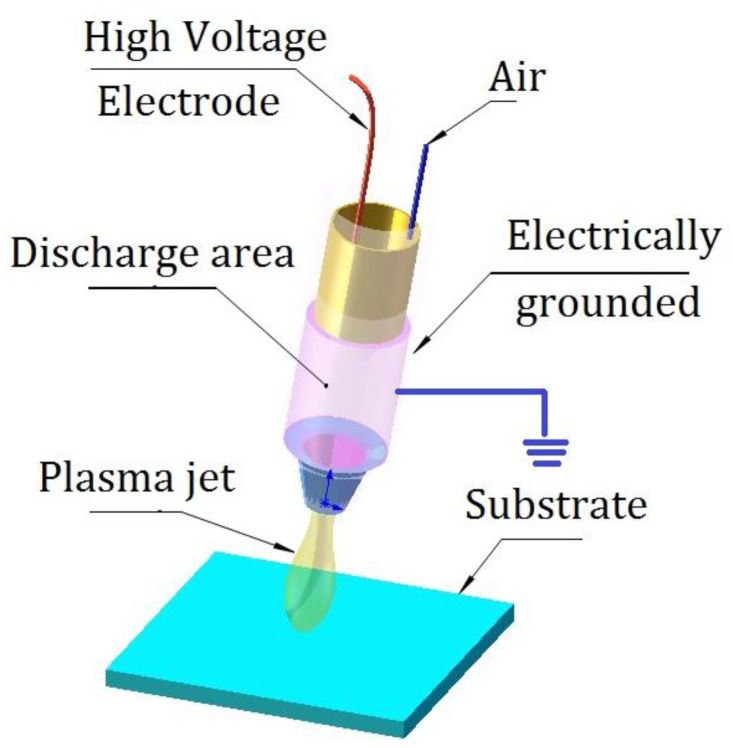
Schematic of a non-thermal atmospheric jet plasma pre-treatment glass substrate.

**Figure 2 materials-16-05152-f002:**
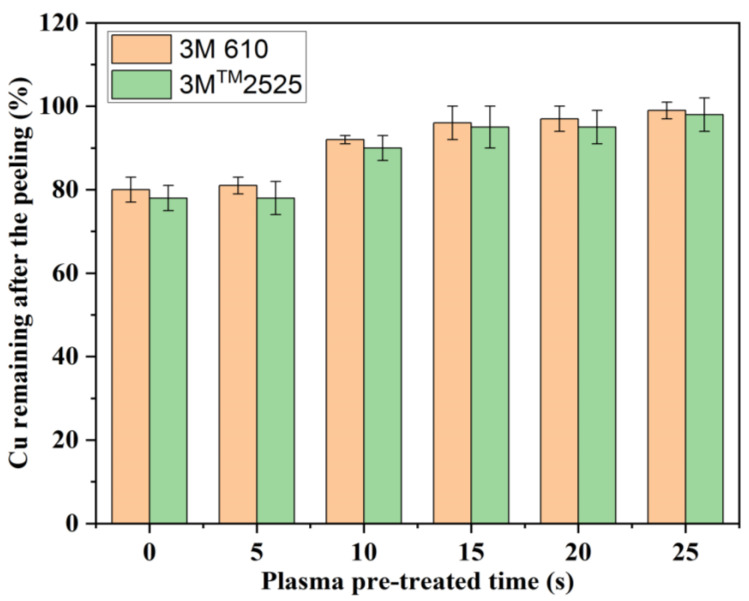
The copper percentage remaining on the glass substrate after the peeling test. (The experimental deviation represented spread over one sample).

**Figure 3 materials-16-05152-f003:**
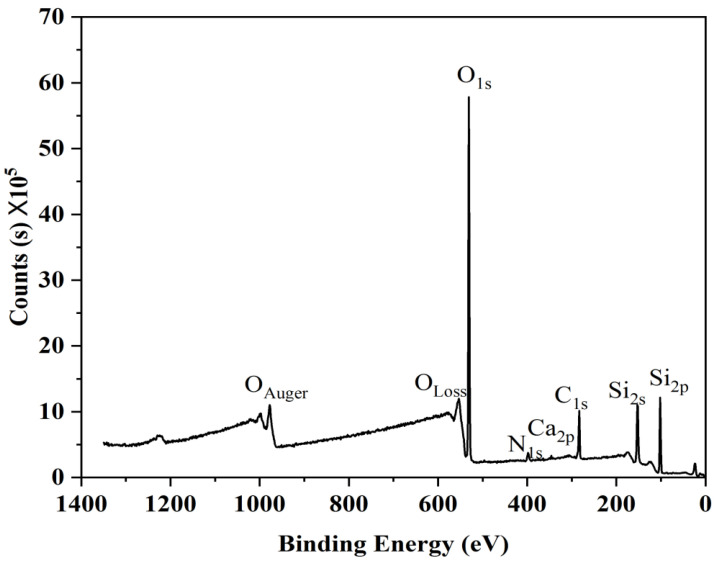
XPS survey spectrum of the glass after 25 s plasma pre-treatment.

**Figure 4 materials-16-05152-f004:**
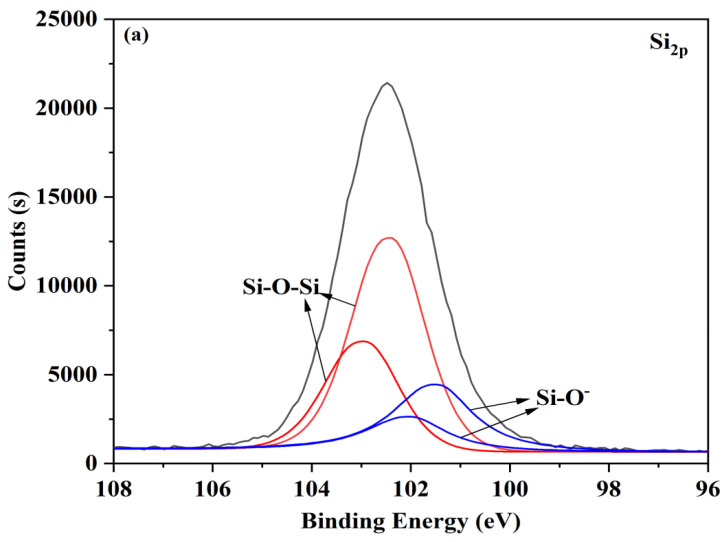
Si2p photoelectron spectra of (**a**) untreated surface and (**b**) plasma treating for 25 s.

**Figure 5 materials-16-05152-f005:**
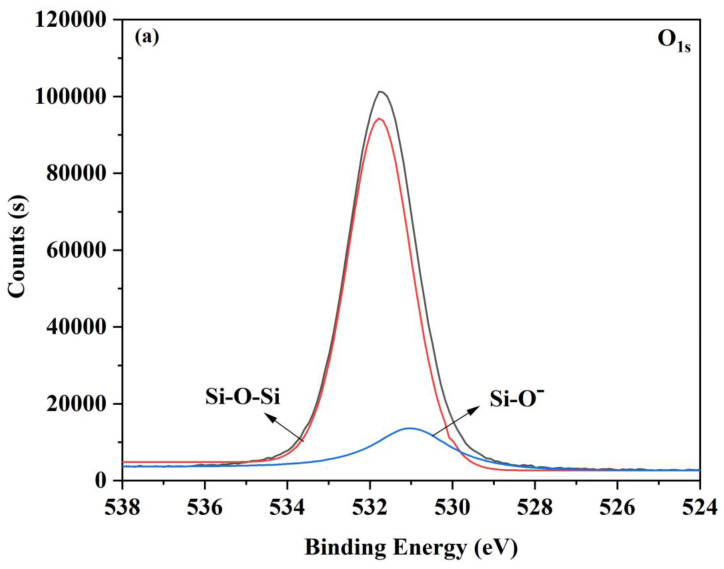
O1s photoelectron spectra of (**a**) untreated surface and (**b**) plasma treating for 25 s.

**Figure 6 materials-16-05152-f006:**
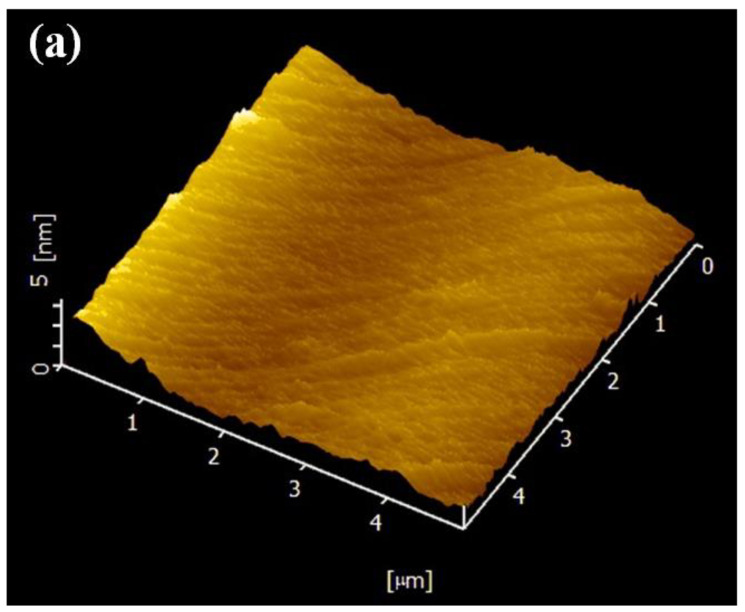
AFM images of (**a**) untreated fused silica glass substrate, (**b**–**f**) fused silica glass substrates treated after 5 s, 10 s, 15 s, 20 s, 25 s air plasma.

**Figure 7 materials-16-05152-f007:**
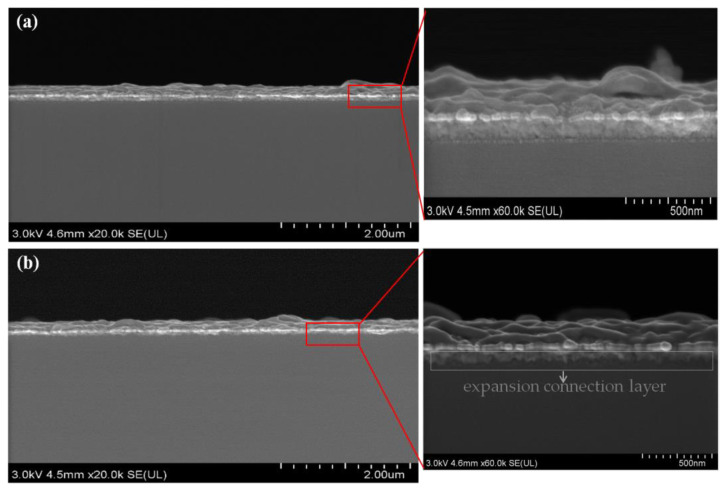
SEM images of the interface of glass substrate and copper layer; (**a**) untreated fused silica glass substrate, (**b**) 25 s air plasma pre-treated glass substrate.

**Table 1 materials-16-05152-t001:** A summary of the adhesion characteristics of copper films on glass substrates.

Copper Films on the Glass		Images after Adhesion Test on Glass on 3M^TM^-2525 Tape	Adhesion Grade (ASTM Class)
Untreated substrate		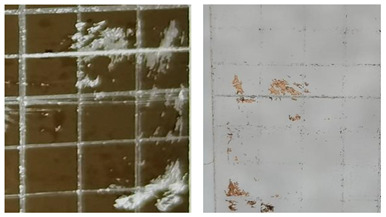	2B
Plasma pre-treated substrates	10 s	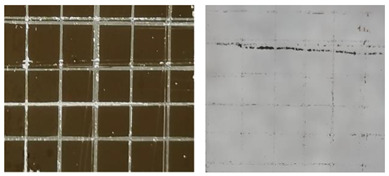	3B
	25 s	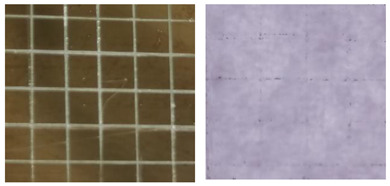	4B

**Table 2 materials-16-05152-t002:** Percentages of Si-N, Si2p and O1s in the XPS peak analysis.

Peak	Si-N	Si2p			O1s		
		Si-O-	Si-O-Si	Total Si2p	Si-O-	Si-O-Si	Total O1s
Untreated	0.33%	2.31%	5.18%	7.49%	2.01%	13.95%	15.96%
Treated 5 s	0.35%	2.27%	4.72%	5.99%	2.12%	15.57%	17.69%
Treated 10 s	0.44%	2.12%	4.00%	6.12%	2.63%	16.74%	19.37%
Treated 15 s	0.47%	1.87%	4.08%	5.95%	2.44%	17.66%	20.10%
Treated 20 s	0.48%	1.54%	3.63%	5.17%	3.15%	17.08%	20.23%
Treated 25 s	0.52%	1.15%	3.66%	4.01%	3.96%	16.95%	20.91%

**Table 3 materials-16-05152-t003:** Contact angles and surface energy of glass samples by plasma pre-treatment.

Samples		Contact Angles		Surface Energy (mJ·m−2)		
		*θ_H_2_O_* (°)	*θ_CH_2_I_2__* (°)	Polar Component	Dispersive Component	Total Surface Energy
Untreated substrate	0 s	70.2 ± 0.5	64.9 ± 0.2	16.1 ± 0.3	18.5 ± 0.2	34.6 ± 0.3
Plasma pre-treated substrates	5 s	60.5 ± 0.2	59.0 ± 0.1	21.8 ± 0.2	20.2 ± 0.1	42.0 ± 0.3
	10 s	59.7 ± 0.4	62.1 ± 0.3	23.7 ± 0.5	18.3 ± 0.2	42.0 ± 0.2
	15 s	53.3 ± 0.4	61.2 ± 0.1	29.0 ± 0.4	17.9 ± 0.3	46.8 ± 0.1
	20 s	48.7 ± 0.3	59.5 ± 0.2	32.4 ± 0.5	18.0 ± 0.2	50.3 ± 0.3
	25 s	41.2 ± 0.9	60.1 ± 0.2	39.2 ± 0.3	16.7 ± 0.1	55.9 ± 0.4

**Table 4 materials-16-05152-t004:** Surface roughness of glass samples as a function of plasma pre-treatment conditions.

Samples	Plasma Treating Time (s)	Ra Roughness (nm)	RSM Roughness (nm)	Adhesion Expressed in Percentage of Copper Remaining on Glass Surface after Peeling (%)		Surface Energy (mJ·m^−2^)	
				3M-610	3M^TM^-2525		
a	0	0.8 ± 0.1	1.0 ± 0.1	80	78		34.6 ± 0.3
b	5	1.0 ± 0.4	1.5 ± 0.2	81	78		42.0 ± 0.3
c	10	1.4 ± 0.1	1.9 ± 0.3	92	90		42.0 ± 0.2
d	15	1.6 ± 0.1	2.2 ± 0.4	96	95		46.8 ± 0.1
e	20	2.2 ± 0.2	3.0 ± 0.5	97	95		50.3 ± 0.3
f	25	2.4 ± 0.3	3. 5 ± 0.1	99	98		55.9 ± 0.4

## Data Availability

The data presented in this study are available upon request from the corresponding author.

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
