# Peer review of "Enhanced Adhesion of Copper Films on Fused Silica Glass Substrate by Plasma Pre-Treatment"

_materials, 2023, doi:10.3390/ma16145152_

Round 1

Reviewer 1 Report

1.      Abstract (page1, line 19): Replace ‘surface roughness (Ra)’ by ‘average surface roughness (Sa)’. Also add spread of roughness values.

2.       Abstract (page1, line 24): Replace ‘with untreated the glass substrate’ by ‘with the untreated glass substrate’ Also, remove extra point.

3.       Introduction: The introduction is well written and provides a good summary of the background on existing know-how. However, the added value and novelty of this work is not highlighted as a small descriptive paragraph is given at the end of introduction. For example, you should mention why you used a non-thermal atmospheric treatment etc.

4.      Materials and methods (page 2, lines 67-70): What was the initial roughness of the substrate?

5.      Figure 1: If this schematic is taken from a book, then you should add reference.

6.      Materials and methods (page 2, lines 79-86): It is ASTM D3359.

7.      Materials and methods (page 2, lines 79-86): Number of repeats for adhesion tests is missing.

8.      Materials and methods: Adhesion tests should be placed after deposition and characterization.

9.      Table 1: Some of the images are blurry.

10.   Results (page 3, lines 118-121): Sentence ‘The photographs that the microscope took of the glass and tapes during the adhesion experiment demonstrated the process, on which morphological changes of copper grids on glass substrates and the number of copper films adhered to the tape were presented clearly’ needs rewriting. Sentences should be shorter and to the point.

11.   Results (page 3, lines 122-123): Almost no copper film adhered to the tape as plasma treatment for 25 s. Almost no copper film adhered to the tape when the copper film was plasma treated for 25 s. 

12.   Results: You do not explain how you quantified adhesion. Also, repeatability of tests needs to be addressed. Does the experimental deviation (given in Figure 1) represent spread over one or multiple samples?

13.   Results: Influence of structure on adhesion is not explained.

14.   Results (page 8, lines 202-206): You mention that ‘According to the interpretation in terms of adhesion, a rise in the number of polar functional groups such as Si-O- (from 2.01% to 3.96%) and Si-N- (from 0.33%-0.52%) on the glass surface likely caused more interactions with copper atoms. As a result, at the Cu/glass interface, an interfacial fusion layer or interface element diffusion may form’. Do you have any evidence for that?

15.   Figure 7: Higher magnification images are needed in order to see the interface between the substrate and the copper.

16.   Results (page 12, lines 230-206): You mention that ‘The plasma treatment may have increased the surface roughness, which increased the number of ordered copper particles and led to a tendency for an atomic expansion connection layer to form between the copper film and the glass, improving the interface layer's ability to adhere’. For me it is unclear how roughness improves adhesion. You do not present any evidence or references to support your assumption.

17. Results: The discussion is very limited. You performed different techniques, but you do not use or correlate them to explain in depth your results. Thus, this work reads more like a technical report rather than a scientific article.

Moderate editing of English is needed. Some sentences are too long, without correct structuring and there is no conclusion made. For example see point 10. 

Reviewer 2 Report

This study reports a systematic study to evaluate the adhesion of Cu films on fused silica glass substrate. The authors report an effective improvement on the adhesion between Cu film and glass substrate by using a plasma pre-treatment. The manuscript should be revised to address the comments below.

1.       English writing. Scientific writing must be carefully revised for accuracy and clarity throughout the manuscript.

2.       Abstract. One example of not clear statement is (L. 19-20) “After 25 s plasma treatment, the surface roughness (Ra) grew from 0.83 nm to 2.43 nm due to increasing number of ordered copper particles on fused silica glass surface,”. The surface roughness is due to the increasing number of ordered copper particles on glass surface? or the surface roughness of glass substrate increased after plasma pre-treatment? Plasma treatment or plasma pre-treatment? Cu or copper?

3.  Introduction. The authors must strengthen the merit of this study by conducting a more detailed analysis of the literature. The authors must clarify the novelty of their study. 

4.       Materials and methods

-The deposition method of Cu films is contradictory. (L. 67-68) “fused silica glasses were employed as the substrates for heat evaporation plating of Cu thin films.” (L. 87-88) “After plasma pre-treatment, copper thin films were deposited by sputtering a copper target (99.999% purity, 30 mm diameter) in argon environment.

- One example of not clear sentence is.  (L. 81-83) “To assess the adhesion properties between the deposited films and 200 nm thick Cu films, 3MTM2525 and 3M 610 Scotch tapes with adhesion strengths on steel of 75 N/mm and 47 N/mm, respectively, were utilized as the adhesive tapes for the test.

- Please describe in detail how the film thickness was determined.

5.       Results

-          (L. 230-231) “The plasma treatment may have increased the surface roughness, which increased the number of ordered copper particles”. Please provide clear evidence on the increased number of ordered Cu particles. Please clarify the meaning of “ordered particles”. The Cu films consist of “ordered” and “disordered” particles? Was the number of each type counted?

-    SEM images. (L. 234-236) “In Figure 7(a), the interface between the glass substrate and the copper layer is clearly defined, and in Figure 7(b), the interface between the two is clearly integrated”. Since it is very difficult to observe a “clear” or “integrated” interface between the glass substrate and Cu films, please provide SEM images in which these features can be clearly observed.

6.       Conclusions. Please check the consistency of L. 246-247 and be concise.

7.       References.  Please check the numbering of references.

 English writing. Scientific writing must be carefully revised for accuracy and clarity throughout the manuscript.

Reviewer 3 Report

Comments:

1)     The authors have mentioned in their manuscript that they are using a non-thermal atmospheric jet plasma to treat their substrate. Non-thermal atmospheric jet plasma typically has a small nozzle tip (a few mm size). Are the authors using some special version that gave a bigger plasma spread over a larger area? As the substrates which they use is of size 20 mm x 10 mm size, can the authors share some insight on how their non-thermal atmospheric jet plasma is applied uniformly in such a short time onto the fused silica glass?

2)     In their adhesion test, for the ASTM 3359, it is common that 3M 610 Scotch tapes is used. But the authors have included another more adhesive tape 3M 2525 for their test and this 3M 2525 result is shown in Table 1A. Table 1A adhesive grade does not seem tally with what is written in the manuscript. If the untreated substrate has more peel off, the ASTM grade measured is 3B. The plasma treated substrate has less peel off, but the ASTM grade become 1B (35-65% lattice detached, even more peel off). And there is no ASTM grade for the 25 s plasma.

3)     It is indeed that the plasma treatment can increase the surface roughness as shown by their AFM result. However, the authors also mentioned the plasma treatment increase the number of ordered copper particles and led to a tendency for an atomic expansion connection layer to form between the copper film and the glass, improving the interface layer’s ability to adhere. Can the authors shown a more zoom in SEM image to support this finding?

Minor comments:

1)     There is typo on the y-axis of figure 2. Cu “ramaining” after the peeling.

2)     There is typo in the abstract on line 18 and line 264, the total surface energy reported in the paragraph is different from the table. So, which is the correct data?

Reviewer 4 Report

The authors enhanced adhesion of Cu films on fused silica by plasma pre-treatment. This method does not require the insertion of adhesion layer. Therefore, the fabrication process becomes simple. This attracts a lot of researchers and companies. However, there are some concerns about the characterization and discussion. If the authors appropriately address my concerns, this study will meet the criteria for the publication in Materials.

Comment list

Comment 1: Poor adhesion of electrode is a big problem in various materials. Therefore, plasma treatment is intriguing a lot of researchers. The authors mentioned the pre-deposition of adhesion layer as the competitive technique. However, in other materials, the electrode film with high adhesion strength was formed by intermixing between film and substrate, such as silicide (Microelectronic engineering 86, 1718 (2009)., Appl. Phys. Lett. 116, 181601 (2020).). By reference to these studies, this study will attract more researchers.

Comment 2: How did the authors measure surface energy?

Comment 3: Probably, surface energy related to adhesion depends on surface roughness. Please clearly describe the reason why adhesion of copper films is increased by plasma treatment. Is the increase of surface roughness the physical origin to enhance the adhesion?

Round 2

Reviewer 1 Report

Dear authors,

After reading the updated version of the article ad your point-by-point reply to comments, I  now believe that the article is appropriate for publication.

Author Response

No response required.

Reviewer 2 Report

The manuscript has been improved, however there are some important remaining issues that need to be addressed.

1.English writing should be clear, concise, and accurate. The entire text must be carefully revised and grammatical and spelling mistakes corrected.

Example: (L.16) “the attention theta Lite optical tensiometer”

2. As plasma pre-treatment of substrates has been a technology previously studied, I can´t find the merit of this study. What is the novelty of this study?

3. Abstract

- (L23-24) The electrostatic force and the mechanical bite force on the interface helped to form an atomic diffusion connection layer and improved interactions between the copper film and the glass substrate. 

- Please provide convincing evidence that an atomic diffusion layer was formed.

- Please define clearly the term “mechanical bite force”.

4. Materials and Methods

- How the film thickness was determined?  Please check L. 112-115 and describe the appropriate procedure.

- What is the meaning of L. 81 “The samples were produced by classical optical manufacturing techniques.”

5. Results

- (L. 139-140) “Almost no copper film adhered to the tape when the copper film was plasma treated for 25 s.” Was the cooper film treated by plasma?

- What is the meaning of L. 156 “(* The experimental deviation represented spread over one sample)”

- Figure 7. Despite higher magnification SEM images have been provided, no difference between the interfaces of glass substrate and the copper film is distinguished. Please provide SEM images in which difference described in the text can be clearly distinguished.

8. Conclusions. Please be concise.

English writing should be clear, concise, and accurate. The entire text must be carefully revised and grammatical and spelling mistakes corrected.

Reviewer 4 Report

Everything was cleared. This manuscript is worth publishing.

Author Response

No response required.